# Effects of the Prophylactic HPV Vaccines on HPV Type Prevalence and Cervical Pathology

**DOI:** 10.3390/v14040757

**Published:** 2022-04-05

**Authors:** Ian N. Hampson

**Affiliations:** Division of Cancer Sciences, University of Manchester, Oxford Rd, Manchester M13 9WL, UK; ian.hampson@manchester.ac.uk; Tel.: +44-7500900871

**Keywords:** HPV, vaccines, prophylactic, cervical cancer, HPV type-replacement, cervical intraepithelial neoplasia, CIN, superinfection exclusion

## Abstract

Vaccination programs with the current prophylactic HPV vaccines started in most countries around 2008 with introduction of the bivalent Cervarix HPV16/18 vaccine, rapidly followed by Gardasil (HPV6/11/16/18) and, finally, Gardasil 9 (HPV6/11/16/18/31/33/45/52/58), from 2015. Many studies have now confirmed their ability to prevent infection with vaccine-covered HPV types, and the subsequent development of either genital warts and/or cervical neoplasia, although this is clearly more effective in younger women vaccinated prior to sexual debut. Most notably, reductions in the prevalence of vaccine-covered HPV types were also observed in unvaccinated women at the same geographical location, presumably by sexual dissemination of these changes, between vaccinated and unvaccinated women. Furthermore, there are several studies that have demonstrated vaccine-associated HPV type-replacement, where vaccine-covered, high-risk HPV types are replaced by high-risk HPV types not covered by the vaccines, and these changes were also observed in vaccinated and unvaccinated women in the same study population. In light of these observations, it is not entirely clear what effects vaccine-associated HPV type-replacement will have, particularly in older, unvaccinated women.

## 1. Introduction

There are now many reports that indicate that prophylactic vaccination with Cervarix or Gardasil, prior to sexual debut and HPV infection, is most effective at preventing both infection and the subsequent development of cervical intraepithelial neoplasia (CIN), specifically caused by the HPV types covered by each vaccine [1]. Indeed, a recent study in Costa Rica confirmed this for the bivalent Cervarix HPV 16/18 vaccine [2], although it is very clear that disease-causing, non-HPV16/18 types are still present in women vaccinated with Cervarix at the same geographical location [3]. Moreover, in spite of the large body of work, which strongly supports the use of Cervarix and Gardasil, there are still some concerns related to adverse events associated with their use. For example, it has been reported that temporal proximity of HPV vaccination, to any type of coincident infection, is associated with increased risk of chronic fatigue syndrome [4]. Furthermore, very recent work has also indicated that undiagnosed and pre-existing mast cell activation syndrome maybe worsened by HPV vaccination [5]. However, these are associative studies, based on a small number of cases, and they do not establish causality.

With respect to vaccine efficacy, it is noteworthy that there are >50 HPV types known to infect genital epithelium, where ~14 are high-risk (HR) and the others are either probable high- or low-risk (LR). Moreover, established HPV infections of one type can influence susceptibility to infection with others by stimulating cross-type immunity [6] and superinfection exclusion [7]. This indicates that the cervix may host a sexually-transmitted, variable meta-community of different HPV types. Given this level of complexity, it is not entirely clear what effect the current HPV type-specific vaccines will have on non-vaccine covered HPV types and associated dysplasia, which has raised some fundamental questions: Are vaccine-covered high-risk HPV types replaced by non-vaccine high-risk HPV types in vaccinated women, and how does this translate over time, with respect to increased risk of developing CIN and cancer? Will changes in HPV types and disease prevalence observed in vaccinated women spread by sexual transmission to unvaccinated women in the same geographical location?

## 2. Post-Vaccination Changes in HPV Type Prevalence

The answers to the preceding questions are not straightforward, since natural cross-type HPV immunity interacts with vaccine-induced, cross-type protection [6]. When the latter is weak, it has been speculated this may increase the time required for HPV type-replacement to occur, which indicates it may be still too early for this to be observed [6]. In spite of this, vaccine-related HPV type-replacement was recently reported in a cross-sectional study of 45,363 women from two autonomous Spanish communities. The prevalence of 35 HPV types was analysed both pre-vaccination (2002–2007) and post-vaccination with Cervarix or Gardasil (2008–2016) [8]. LR HPV6/11 infections showed a significant post-vaccination decline and, although HPV16 also dropped over this period, it did not achieve statistical significance, whereas HPV18 prevalence showed no change. Most notably, HR types 31, 52, and 45, which are not covered by Gardasil, all showed a significant post-vaccination increase in prevalence, which clearly supports HPV type-replacement. Furthermore, although types 31, 52, and 45 are covered by Gardasil 9, it is significant that HR HPV types 35, 39, 56, 59, and 68 are not covered by this vaccine, and they also increased in prevalence in the post-vaccination population (see Supplementary Material, additional data file, in [8]). These observations are consistent with a recent study that assessed HPV infection rates in vaccinated adolescent and young adult women, between 2008 and 2019, at a New York City adolescent-specific health centre [9]. These authors found that, although the incidence of vaccine-covered HPV types declined during this period, other non-vaccine types increased (Figures 1 and 2 in [9]). Indeed, a similar study was conducted at a youth clinic in Stockholm, which also showed a significant reduction of vaccine-covered HPV types, in both vaccinated and unvaccinated women, between 2008 and 2018 [10]. However, it was also found that non-vaccine covered high-risk HPV types increased significantly over this period, again in both vaccinated and unvaccinated women (See Figure 1 in [10]). The observed transfer of vaccine-induced changes in HPV type prevalence from vaccinated to unvaccinated women is concerning, since it is difficult to predict what effects the increased prevalence of non-vaccine covered HR HPV types will have in unvaccinated women. Curiously, in spite of the extensive national vaccination program, Sweden is currently experiencing an ongoing increase in the incidence of cervical cancer [11].

## 3. Post-Vaccination Changes in HPV Type Prevalence Associated with Cervical Dysplasia

The findings discussed, thus far, have analysed the effects of vaccination on HPV type prevalence, irrespective of cervical dysplasia, and it is very clear that this should also be evaluated. In this regard, a very recent study carried out in Northern Italy assessed the HPV type and CIN status of 5807 women, aged between 21–65 years, with abnormal pap smears, who attended for colposcopy over a 15-year period, between 2005 and 2019 [12]. Analysis of the 3475 women who had a colposcopy-directed biopsy showed a time-dependent reduction in the incidence of HPV 16 and 31 in women aged 21–29 years diagnosed with CIN1. However, this was not seen in women over 30 years old, and no reduction in HPV16 incidence was observed in women with CIN2. Furthermore, there was also an equivalent increase in the detection of non-vaccine covered HPVs, in addition to lesions which were either HPV negative or tested positive for unknown HPV types across all age groups. This study clearly supports vaccine-related HPV type-replacement; most significantly, the prevalence of all seven HR-HPVs targeted by Gardasil 9 remained unchanged, irrespective of age or severity of cervical lesion. These results are consistent with a study carried out in five US States, from 2008–2015, which analysed the incidence of 16,572 CIN2 lesions in women aged 18–39 years [13]. In those who attended for screening, a reduced incidence of CIN2 was observed over time in women aged 18–24 years, whereas women aged 25–39 years showed a marked increase. Furthermore, the same trend was observed for CIN3. Japan initiated an HPV vaccination program in 2010; although reinstated in 2021, it was discontinued in 2013, due to concerns over adverse events. The OCEAN (Osaka clinical research of HPV vaccine) study evaluated the effects of vaccination, between 2010 and 2015, in a cohort of 2814 women, aged between 12 and 18 years [14]. Of these, 170 women were tested at the age of 20–21 years for cervical cytology/pathology and HPV type and compared to an unvaccinated cohort of 877 women from the same age group and geographical location. Although only a small study, a decrease in the overall prevalence of some HR HPV types, most notably HPV16/18, was observed; yet, consistent with other studies, an increase in the prevalence of HR types 56 and 35 was also seen in vaccinated vs unvaccinated women. Comparison of cytology and pathology between these groups showed a modest increase in low-grade CIN1 lesions in vaccinated women, whereas no high-grade CIN2/3 lesions were found in this group, compared to four CIN2′s detected in the unvaccinated cohort. This work clearly supports the efficacy of the HPV vaccines against vaccine-covered HPV types, combined with some cross-protection against non-vaccine types. However, even over the short time period of this study, it also provides some evidence for vaccination-associated, HPV type-replacement and does not address any potential longer-term impacts in older, unvaccinated women.

## 4. Post-Vaccination Changes in the Incidence of Cervical Cancer

Since cervical cancer usually takes ~10 years to develop, the aforementioned studies analysed the effects of the vaccines on HPV type prevalence and the incidence of CIN as proxies for the subsequent development of cancer. In spite of this, it would be expected that the incidence of invasive disease would at least remain stable or show a moderate decline post-vaccination. However, using the United States Cancer Statistics (USCS) database, a study on cervical cancer incidence was carried on 15–29 year old women in the US, between 1999–2017 [15]. Although this showed a reduction in incidence for women aged 15–24 years, from 2012–2017, women aged 25–29 showed an increase during the same period. Furthermore, it is notable that, prior to the start of vaccination in 2008, the overall incidence of cervical cancer in the younger age group in the US was very low, at 90 cases, and increased 4 fold in the 25–29 year age group to 363 cases. Even so, these figures are much lower than the 1663 cases diagnosed in women aged 30–39 years during the same period [16].

The current trend of increased incidence of cervical cancer in Sweden [11] has already been discussed with both Norway and Finland also reporting an increase during the post-vaccination period ([17] accessed 28 February 2022). Moreover, an increase has also been observed in the UK in older women, aged 25–40 years ([18] accessed 28 February 2022).

## 5. Potential Economic Implications

Cost-effectiveness analyses (CEA), carried out on the current HPV vaccines, is dependent on the length of protection, degree of cross-type protection, and extent of any HPV type-replacement observed [19]. Thus, it is very clear that the previously discussed observations may have a significant impact on CEA. Most notably, the finding of the indirect dissemination of vaccine-related changes in HPV type prevalence between vaccinated and older unvaccinated women at the same geographical location has yet to be evaluated. Indeed, the cost estimates of each quality-adjusted life year (QALY) gained from vaccination, have been shown to increase markedly for catch-up vaccinated adults aged >30 years [20]. In light of these observations, it is uncertain how this will impact vaccine use in developing nations, since, when resources are limited, any factors which may compromise overall vaccine efficacy (and, thereby, influence CEA) have to be carefully considered.

## 6. Conclusions

Collectively, the aforementioned studies clearly demonstrate vaccine-related direct and indirect herd effects, where changes in the incidence of vaccine- and non-vaccine-covered HPV infections were transferred between vaccinated and unvaccinated women over time. Most significantly, it will be important to carefully monitor the potential impact of these alterations in older unvaccinated women, where there is no vaccine-related cross-type protection, and there may be increased potential for iatrogenic outcomes.

## Data Availability

Not applicable.

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
