# Peer review of "Effects of the Prophylactic HPV Vaccines on HPV Type Prevalence and Cervical Pathology"

_viruses, 2022, doi:10.3390/v14040757_

Round 1

Reviewer 1 Report

The author provides a comprehensive review on an important issue concerning the possible replacement of vaccine-covered hrHPV genotypes by non-vaccine hrtypes in vaccinated women including subsequent clinical consequences. In addition, the manuscript discusses the effect of changes in HPV types and disease prevalence on unvaccinated women that live in the same environment. 

The review is brief and focuses on selected studies relevant for the topics. I propose the acceptance of the manuscript with the following revision: 

  1. In the title, the word "unexpected" is, in my opinion, not the best choice and does not fit the text. Please change the title to a more descriptive and specific one (It does not have to be short, the important thing is to revfect the content of the review.
  2. It would be interesting to include some very recent studies on vaccine efficacy from non-european cohorts to re-enforce the evidence on the reduction or dissappearance of HIV-16/18 in vaccinated women (for example Hiramatsu et al, Hum Vaccin Immunother, 2022, the OCEAN study from Japan or any other the author feels are appropriate)
  3. Taking into consideration the data presented in sections 2, 3 and 4, it would be interesting to add a sentence or two describing the opinion of the author on the cost-effectiveness analysses of vaccines in the context of data on genotype replacement and cross-protection (perhaps as a comment on Pike et al, Appl Health Econ Health Policy, 2022. 
  4. The majority of studies reviewed in the manuscript are from Europe and USA. It would be interesting to commend on the abovementioned issues in the developing world. 

Author Response

Response to Reviewer 1

(1) I have done as the referee suggests and changed the title to make it more definitive. It now reads "Effects of the Prophylactic HPV Vaccines on HPV Type Prevalence and Cervical Pathology".

(2) I thank the referee for this suggestion and have added the Hiramatsu et al 2022 reference (Ref 15) plus a discussion of this article in Section 3.

(3) This is a very worthwhile suggestion which had not occurred to me and I am grateful to the referee for this observation. A short discussion of the potential impact of genotype replacement and cross-protection on vaccine cost effectiveness analysis has now been added as a new Section 5 entitled “Potential Economic Implications” .

(4) A short statement on the potential impact of the issues discussed on vaccine usage in developing nations has been added in the new Section 5.   

Reviewer 2 Report

Hampson aimed to demonstrate, in a perspective article, vaccine-related direct and indirect herd-effects where changes in the incidence of vaccine-covered and non-vaccine-covered HPV infections were transferred between vaccinated and unvaccinated women over time.

The paper contains an interesting topic, showing that prophylactic HPV vaccine effects is a trend in keeps growing. This reviewer suggest to accept the manuscript in current form.

Author Response

Response to Review 2

I thank would like to thank the reviewer for their recomendation.

Reviewer 3 Report

The author in the manuscript provides a perspective on supposed effects associated with prophylactic use of the HPV vaccine. In my opinion, the perspective is flawed by several biases and carries a number of assumptions that are not adequately supported by references. To me, this makes the conclusions drawn from the perspective ambiguous and in some ways dangerous, as it appears to diminish the value of HPV vaccination as a key public health tool.

Please find my comments below:

Unanticipated Effects of the Prophylactic HPV Vaccines

Abstract: Vaccination programs with the current prophylactic HPV vaccines started in most countries around 2008 with introduction of the bivalent Cervarix HPV16/18 vaccine rapidly followed by Gardasil (HPV6/11/16/18) and finally Gardasil 9 (HPV6/11/16/18/ 31/33/45/52/58) from 2015. Many studies have now confirmed their ability to prevent infection with vaccine-covered HPV types and subsequent development of either genital warts and/or cervical neoplasia although this is clearly more effective in younger women vaccinated prior to sexual debut. Most notably, reductions in the prevalence of vaccine-covered HPV types were also observed in unvaccinated women at the same geographical location, presumably by sexual dissemination of these changes between vaccinated and unvaccinated women. Furthermore, there are several studies which have demonstrated vaccine-associated HPV type-replacement where vaccine-covered high-risk HPV types are replaced by high-risk HPV types not covered by the vaccines and these changes were also observed in vaccinated and unvaccinated women in the same study population. In light of these observations, it is not entirely clear what effects vaccine-associated HPV type- replacement will have, particularly in older unvaccinated women.

The title is not consistent with the content of the abstract and the entire manuscript. In fact, it speaks of "unanticipated effects of prophylactic HPV vaccination", referring only to the female population. Not considering all HPV-related pathologies (head and neck cancer, anal cancer etc.) and therefore not extending the reflection to the use of HPV vaccine in male subjects is anachronistic. Gender-neutral campaigns have been running for years in many countries. Moreover, key populations of the vaccine offer, such as MSM, are excluded.

Please find some references below:

  • Hartwig S, St Guily JL, Dominiak-Felden G, Alemany L, de Sanjosé S. Estimation of the overall burden of cancers, precancerous lesions, and genital warts attributable to 9-valent HPV vaccine types in women and men in Europe. Infect Agent Cancer. 2017;12:19. Published 2017 Apr 11. doi:10.1186/s13027-017-0129-6 (https://www.ncbi.nlm.nih.gov/pmc/articles/PMC5387299/)
  • Elfström KM, Lazzarato F, Franceschi S, Dillner J, Baussano I. Human Papillomavirus Vaccination of Boys and Extended Catch-up Vaccination: Effects on the Resilience of Programs. J Infect Dis. 2016 Jan 15;213(2):199-205. doi: 10.1093/infdis/jiv368. Epub 2015 Jul 3. PMID: 26142436 (https://pubmed.ncbi.nlm.nih.gov/26142436/)
  • Efficacy of the quadrivalent hpv vaccine against anal infections and anal intraepithelial neoplasia (ain) (palefsky jm et al, 2011) October 27, 2011 N Engl J Med 2011; 365:1576-1585
    DOI: 10.1056/NEJMoa1010971 (https://www.nejm.org/doi/full/10.1056/nejmoa1010971)
  • Olsen J, Jørgensen TR. Revisiting the cost-effectiveness of universal HPV-vaccination in Denmark accounting for all potentially vaccine preventable HPV-related diseases in males and females. Cost Eff Resour Alloc. 2015 Feb 11;13:4. doi: 10.1186/s12962-015-0029-9. PMID: 25694771; PMCID: PMC4331443. (https://pubmed.ncbi.nlm.nih.gov/25694771/ )

 Moreover, in spite of the large body of work which strongly supports the use of Cervarix and Gardasil, there are still some concerns related to adverse events associated with their use. For example, it has been reported that temporal proximity of HPV vaccination to any type of coincident infection, is associated with increased risk of chronic fatigue syndrome [4]. Furthermore, very recent work has also indicated that undiagnosed and pre-existing mast cell activation syndrome maybe worsened by HPV vaccination [5].

Evidence shows that the HPV vaccines are effective and safe. The side effects mentioned by the author (chronic fatigue syndrome, etc.) have been disproved.

It should also be remembered that the basic flaw in statements of this kind is that associations should not be confused with causation (correlation is not causation).

Please find some references below:

  • Scheller NM, Svanström H, Pasternak B, Arnheim-Dahlström L, Sundström K, Fink K, Hviid A. Quadrivalent HPV vaccination and risk of multiple sclerosis and other demyelinating diseases of the central nervous system. JAMA. 2015 Jan 6;313(1):54-61. doi: 10.1001/jama.2014.16946. PMID: 25562266. (https://pubmed.ncbi.nlm.nih.gov/25562266/)
  • https://www.ema.europa.eu/en/documents/referral/hpv-vaccines-article-20-procedure-ema-confirms-evidence-does-not-support-they-cause-crps-pots_en.pdf
  • https://www.nap.edu/catalog/13164/adverse-effects-of-vaccines-evidence-and-causality
  • Willame C, Gadroen K, Bramer W, Weibel D, Sturkenboom M. Systematic Review and Meta-analysis of Postlicensure Observational Studies on Human Papillomavirus Vaccination and Autoimmune and Other Rare Adverse Events. Pediatr Infect Dis J. 2020 Apr;39(4):287-293. doi: 10.1097/INF.0000000000002569. PMID: 31876615 (https://pubmed.ncbi.nlm.nih.gov/31876615/ )

Author Response

Response to Review 3

I would like to thank the reviewer for their suggestions to which I have responded as follows:

Reviewer: In my opinion, the perspective is flawed by several biases and carries a number of assumptions that are not adequately supported by references. To me, this makes the conclusions drawn from the perspective ambiguous and in some ways dangerous, as it appears to diminish the value of HPV vaccination as a key public health tool.

Response. It was not my intention to diminish the value of the HPV vaccines and I have tried to present a balanced perspective which highlights both positive and negative aspects together with appropriate references. In my defence, both the first two references cited in the Introduction strongly support efficacy of the vaccines against vaccine-covered HPV types and associated pathology.   

Reviewer: The title is not consistent with the content of the abstract and the entire manuscript. In fact, it speaks of "unanticipated effects of prophylactic HPV vaccination", referring only to the female population. Not considering all HPV-related pathologies (head and neck cancer, anal cancer etc.) and therefore not extending the reflection to the use of HPV vaccine in male subjects is anachronistic. Gender-neutral campaigns have been running for years in many countries. Moreover, key populations of the vaccine offer, such as MSM, are excluded.

Response: The reviewer is quite correct about the title and I have changed this to more accurately reflect the scope of the article. It now reads “Effects of the Prophylactic HPV Vaccines on HPV Type Prevalence and Cervical Pathology”.  Regarding the issue of gender neutrality, I deliberately chose to focus exclusively on cervical disease since the vast majority of vaccination programs started with females and the time for development of HPV-related cervical pathologies is generally shorter than for other gender-neutral HPV target sites. Thus, the impact of the vaccines can be assessed more thoroughly over a longer period of time by specifically focusing on cervical pathology.  

Reviewer: Evidence shows that the HPV vaccines are effective and safe. The side effects mentioned by the author (chronic fatigue syndrome, etc.) have been disproved. It should also be remembered that the basic flaw in statements of this kind is that associations should not be confused with causation (correlation is not causation).

Response: I take the reviewers point and I fully understand the issues regarding drawing conclusions about causality. Thus, in order to clarify this, I have inserted the statement; “However, these are associative studies based on a small numbers of cases and they do not establish any causality” at the end of the Introduction.  In my defence, the two references cited are both very recent and were included to provide examples of the work carried out on potential side effects. 

Round 2

Reviewer 3 Report

I thank the author for responding to some of my comments in a comprehensive manner. 
However, I still disagree with some of the statements in the manuscript.